# NOISY MACHINES: UNDERSTANDING NOISY NEURAL NETWORKS AND ENHANCING ROBUSTNESS TO ANALOG HARDWARE ERRORS USING DISTILLATION

## ABSTRACT

The success of deep learning has brought forth a wave of interest in computer hardware design to better meet the high demands of neural network inference. In particular, analog computing hardware has been heavily motivated specifically for accelerating neural networks, based on either electronic, optical or photonic devices, which may well achieve lower power consumption than conventional digital electronics. However, these proposed analog accelerators suffer from the intrinsic noise generated by their physical components, which makes it challenging to achieve high accuracy on deep neural networks. Hence, for successful deployment on analog accelerators, it is essential to be able to train deep neural networks to be robust to random continuous noise in the network weights, which is a somewhat new challenge in machine learning. In this paper, we advance the understanding of noisy neural networks. We outline how a noisy neural network has reduced learning capacity as a result of loss of mutual information between its input and output. To combat this, we propose using knowledge distillation combined with noise injection during training to achieve more noise robust networks, which is demonstrated experimentally across different networks and datasets, including ImageNet. Our method achieves models with as much as $\sim 2\times$ greater noise tolerance compared with the previous best attempts, which is a significant step towards making analog hardware practical for deep learning.

## 1 INTRODUCTION

Deep neural networks (DNNs) have achieved unprecedented performance over a wide variety of tasks such as computer vision, speech recognition, and natural language processing. However, DNN inference is typically very demanding in terms of compute and memory resources. Consequently, larger models are often not well suited for large-scale deployment on edge devices, which typically have meagre performance and power budgets, especially battery powered mobile and IoT devices. To address these issues, the design of specialized hardware for DNN inference has drawn great interest, and is an extremely active area of research. To date, a plethora of techniques have been proposed for designing efficient neural network hardware (Sze et al., 2017).

In contrast to the current status quo of predominantly digital hardware, there is significant research interest in analog hardware for DNN inference. In this approach, digital values are represented by analog quantities such as electrical voltages or light pulses, and the computation itself (e.g., multiplication and addition) proceeds in the analog domain, before eventually being converted back to digital. Analog accelerators take advantage of particular efficiencies of analog computation in exchange for losing the bit-exact precision of digital. In other words, analog compute is cheap but somewhat imprecise. Analog computation has been demonstrated in the context of DNN inference in both electronic (Binas et al., 2016), photonic (Shen et al., 2017) and optical (Lin et al., 2018) systems. Analog accelerators promise to deliver at least two orders of magnitude better performance over a conventional digital processor for deep learning workloads in both speed (Shen et al., 2017) and energy efficiency (Ni et al., 2017). Electronic analog DNN accelerators are arguably the most mature technology and hence will be our focus in this work.

The most common approach to electronic analog DNN accelerator is *in-memory computing*, which typically uses non-volatile memory (NVM) crossbar arrays to encode the network weights as analog values. The NVM itself can be implemented with memristive devices, such as metal-oxide resistive random-access memory (ReRAM) (Hu et al., 2018) or phase-change memory (PCM) (Le Gallo et al., 2018; Boybat et al., 2018; Ambrogio et al., 2018). The matrix-vector operations computed during inference are then performed in parallel inside the crossbar array, operating on analog quantities for weights and activations. For example, addition of two quantities encoded as electrical currents can be achieved by simply connecting the two wires together, whereby the currents will add linearly according to Kirchhoff's current law. In this case, there is almost zero latency or energy dissipation for this operation.

Similarly, multiplication with a weight can be achieved by programming the NVM cell conductance to the weight value, which is then used to convert an input activation encoded as a voltage into a scaled current, following Ohm's law. Therefore, the analog approach promises significantly improved throughput and energy efficiency. However, the analog nature of the weights makes the compute noisy, which can limit inference accuracy. For example, a simple two-layer fully-connected network with a baseline accuracy of $91.7\%$ on digital hardware, achieves only $76.7\%$ when implemented on an analog photonic array (Shen et al., 2017). This kind of accuracy degradation is not acceptable for most deep learning applications. Therefore, the challenge of imprecise analog hardware motivates us to study and understand *noisy neural networks*, in order to maintain inference accuracy under noisy analog computation.

The question of how to effectively learn and compute with a noisy machine is a long-standing problem of interest in machine learning and computer science (Stevenson et al., 1990; Von Neumann, 1956). In this paper, we study noisy neural networks to understand their inference performance. We also demonstrate how to train a neural network with distillation and noise injection to make it more resilient to computation noise, enabling higher inference accuracy for models deployed on analog hardware. We present empirical results that demonstrate state-of-the-art noise tolerance on multiple datasets, including ImageNet.

The remainder of the paper is organized as follows. Section 2 gives an overview of related work. Section 3 outlines the problem statement. Section 4 presents a more formal analysis of noisy neural networks. Section 5 gives a distillation methodology for training noisy neural networks, with experimental results. Finally, Section 6 provides a brief discussion and Section 7 closes with concluding remarks.

## 2 RELATED WORK

Previous work broadly falls under the following categories: studying the effect of analog computation noise, analysis of noise-injection for DNNs, and use of distillation in model training.

**Analog Computation Noise Models**  In Rekhi et al. (2019), the noise due to analog computation is modeled as additive parameter noise with zero-mean Gaussian distribution. The variance of this Gaussian is a function of the effective number of bits of the output of an analog computation. Similarly, the authors in Joshi et al. (2019) also model analog computation noise as additive Gaussian noise on the parameters, where the variance is proportional to the range of values that their PCM device can represent. Some noise models presented have included a more detailed account of device-level interactions, such as voltage drop across the analog array (Jain et al., 2018; Feinberg et al., 2018), but are beyond the scope of this paper. In this work, we consider an additive Gaussian noise model on the weights, similar to Rekhi et al. (2019); Joshi et al. (2019) and present a novel training method that outperforms the previous work in model noise resilience.

**Noise Injection for Neural Networks**  Several stochastic regularization techniques based on noise-injection and dropout (Srivastava et al., 2014; Noh et al., 2017; Li & Liu, 2016) have been demonstrated to be highly effective at reducing overfitting. For generalized linear models, dropout and additive noise have been shown to be equivalent to adaptive $L_2$ regularization to a first order (Wager et al., 2013). Training networks with Gaussian noise added to the weights or activations can also increase robustness to variety of adversarial attacks (Rakin et al., 2018). Bayesian neural networks replace deterministic weights with distributions in order to optimize over the posterior

distribution of the weights (Kingma & Welling, 2013). Many of these methods use noise injection at *inference time* to approximate weight distribution; in Gal & Ghahramani (2016) a link between Gaussian processes and dropout is established in an effort to model the uncertainty of the output of a network. A theoretical analysis by Stevenson et al. (1990) has shown that for neural networks with adaptive linear neurons, the probability of error of a noisy neural network classifier with weight noise increases with the number of layers, but largely independent of the number of weights per neuron or neurons per layer.

**Distillation in Training**   Knowledge distillation (Hinton et al., 2015) is a well known technique in which the soft labels produced by a *teacher model* are used to train a *student model* which typically has reduced capacity. Distillation has shown merit for improving model performance across a range of scenarios, including student models lacking access to portions of training data (Micaelli & Storkey, 2019), quantized low-precision networks (Polino et al., 2018; Mishra & Marr, 2017), protection against adversarial attacks (Papernot et al., 2016; Goldblum et al., 2019), and in avoiding catastrophic forgetting for multi-task learning (Schwarz et al., 2018). To the best of our knowledge, our work is the first to combine distillation with noise injection in training to enhance model noise robustness.

## 3    PROBLEM STATEMENT

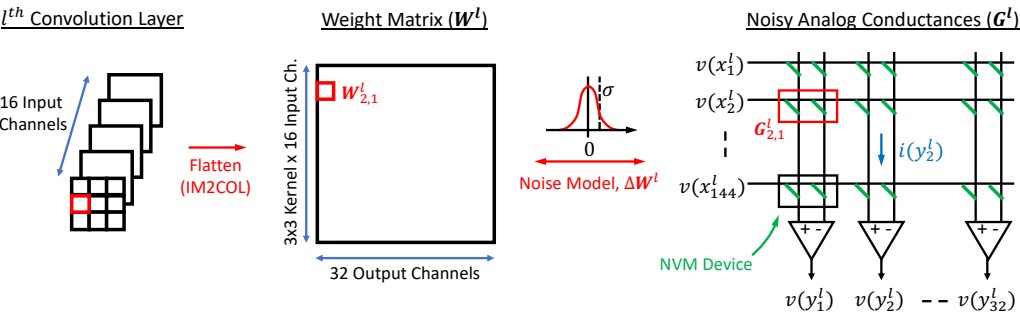

Figure 1: Deploying a neural network layer, $l$, on an analog in-memory crossbar involves first flattening the filters for a given layer into weight matrix $\mathbf{W}^1$, which is then programmed into an array of NVM devices which provide differential conductances $\mathbf{G}^1$ for analog multiplication. A random Gaussian $\Delta\mathbf{W}^1$ is used to model the inherent imprecision in analog computation.

Without loss of generality, we model a general noisy machine after a simple memristive crossbar array, similar to Shafiee et al. (2016). Figure 1 illustrates how an arbitrary neural network layer, $l$, such as a typical $3 \times 3$ convolution, can be mapped to this hardware substrate by first flattening the weights into a single large 2D matrix, $\mathbf{W}^1$, and then programming each element of this matrix into a memristive cell in the crossbar array, which provides the required conductances $\mathbf{G}^1$ (the reciprocal of resistance) to perform analog multiplication following Ohm's law, $i_{out} = v_{in}G$. Note that a pair of differential pair of NVM devices are typically used to represent a *signed* quantity in $\mathbf{G}^1$. Subsequently, input activations, $x^l$ converted into continuous voltages, $v(x^l)$, are streamed into the array rows from the left-hand side. The memristive devices connect row with columns, where the row voltages are converted into currents scaled by the programmed conductance, $\mathbf{G}$, to generate the currents $i(y^l)$, which are differential in order to represent both positive and negative quantites with unipolar signals. The currents from each memristive device essentially add up for free where they are connected in the columns, according to Kirchhoff's current law. Finally, the differential currents are converted to bipolar voltages, $v(y^l)$, which are they digitized before adding bias, and performing batch normalization and ReLU operations, which are not shown in Figure 1.

However, the analog inference hardware of Figure 1 is subject to real-world non-idealities, typically attributed to variations in: 1) manufacturing process, 2) supply voltage and 3) temperature, *PVT variation* collectively, all of which result in noise in the system. Below we discuss the two key components in terms of analog noise modeling.

**Data Converters.** Digital-to-analog converter (DAC) and analog-to-digital converter (ADC) circuits are designed to be robust to PVT variation, but in practice these effects do degrade the *resolution* (i.e. number of bits). Therefore, we consider *effective* number of bits (ENOB), which is a lower bound on resolution in the presence of non-idealities. Hence, we use activation and weight quantization with ENOB data converters and no additional converter noise modeling.

**NVM cells.** Due to their analog nature, memristive NVM cells have limited precision, due to the read and write circuitry (Joshi et al., 2019). In between write and read operations, their stored value is prone to *drift* over time. Long-term drift can be corrected with periodic refresh operations. At shorter timescales, time-varying noise may be encountered. For most of the experiments in this paper, we model generic NVM cell noise as an additive zero-mean *i.i.d.* Gaussian error term on the weights of the model in each particular layer $\Delta \mathbf{W}^l \sim \mathcal{N}(\Delta \mathbf{W}^l; 0, \sigma_{N,l}^2 \mathbf{I})$. This simple model, described more concretely in Section 5, is similar to that used by Joshi et al. (2019) which was verified on real hardware. In addition, we also investigate spatially-varying and time-varying noise models in Section 5.2 (Table 1).

## 4 ANALYSIS OF NOISY NEURAL NETWORKS

### 4.1 BIAS VARIANCE DECOMPOSITION FOR NOISY WEIGHTS

Naively deploying an off-the-shelf pretrained model on a noisy accelerator will yield poor accuracy for a fundamental reason. Consider a neural network $f(\mathbf{W}; \boldsymbol{x})$ with weights $\mathbf{W}$ that maps an input $\boldsymbol{x} \in \mathbb{R}^n$ to an output $y \in \mathbb{R}$. In the framework of statistical learning, $\boldsymbol{x}$ and $y$ are considered to be randomly distributed following a joint probability distribution $p(\boldsymbol{x}, y)$. In a noisy neural network, the weights $\mathbf{W}$ are also randomly distributed, with distribution $p(\mathbf{W})$. The expected Mean Squared Error (MSE) of this noisy neural network can be decomposed as

$$
\begin{aligned}
&\mathbb{E}_{(\boldsymbol{x},y)\sim p(\boldsymbol{x},y), \mathbf{W}\sim p(\mathbf{W})}[(f(\mathbf{W}; \boldsymbol{x}) - y)^2] \\
=& \mathbb{E}_{(\boldsymbol{x},y)\sim p(\boldsymbol{x},y), \mathbf{W}\sim p(\mathbf{W})}[(f(\mathbf{W}; \boldsymbol{x}) - \mathbb{E}_{\mathbf{W}\sim p(\mathbf{W})}[f(\mathbf{W}; \boldsymbol{x})] + \mathbb{E}_{\mathbf{W}\sim p(\mathbf{W})}[f(\mathbf{W}; \boldsymbol{x})] - y)^2] \\
=& \mathbb{E}_{\boldsymbol{x}\sim p(\boldsymbol{x})}[\mathbb{E}_{\mathbf{W}\sim p(\mathbf{W})}[(f(\mathbf{W}; \boldsymbol{x}) - \mathbb{E}_{\mathbf{W}\sim p(\mathbf{W})}[f(\mathbf{W}; \boldsymbol{x})])^2]] \\
& + \mathbb{E}_{(\boldsymbol{x},y)\sim p(\boldsymbol{x},y)}[(\mathbb{E}_{\mathbf{W}\sim p(\mathbf{W})}[f(\mathbf{W}; \boldsymbol{x})] - y)^2].
\end{aligned} \tag{1}
$$

The first term on the right hand side of Equation 1 is a variance loss term due to randomness in the weights and is denoted as $l_{\text{var}}$. The second term is a squared bias loss term which we call $l_{\text{bias}}$. However, typically a model is trained to minimize the empirical version of expected loss $l_{\text{pretrained}} = \mathbb{E}_{(\mathbf{x},y)\sim p(\boldsymbol{x},y)}[(f(\mathbb{E}[\mathbf{W}]; \boldsymbol{x}) - y)^2]$. We assume that the noise is centered such that pretrained weights are equal to $\mathbb{E}[\mathbf{W}]$. A pretrained model is therefore optimized for the wrong loss function when deployed on a noisy accelerator. To show this in a more concrete way, a baseline LeNet model (32 filters in the first convolutional layer, 64 filters in the second convolutional layer and 1024 neurons in the fully-connected layer) (LeCun et al., 1998) is trained on MNIST dataset to 99.19% accuracy and then exposed to Gaussian noise in its weights, numerical values of these loss terms can be estimated. The expected value of the network output $\mathbb{E}_{\mathbf{W}}[f(\mathbf{W}; \boldsymbol{x})]$ is estimated by averaging over outputs of different instances of the network for the same input $\boldsymbol{x}$. We perform inference on $n = 100$ different instances of the network and estimate the loss terms as

$$
\overline{f}(\mathbf{W}; \boldsymbol{x}) = \mathbb{E}_{\mathbf{W}\sim p(\mathbf{W})}[f(\mathbf{W}; \boldsymbol{x})] \simeq \frac{1}{n} \sum_{i=1}^{n} f(\mathbf{W}_i; \boldsymbol{x}), \tag{2}
$$

$$
\hat{l}_{\text{var}} = \frac{1}{N} \sum_{j=1}^{N} \frac{1}{n} \sum_{i=1}^{n} (f(\mathbf{W}_i; \boldsymbol{x}_j) - \overline{f}(\mathbf{W}; \boldsymbol{x}_j))^2, \tag{3}
$$

$$
\hat{l}_{\text{bias}} = \frac{1}{N} \sum_{j=1}^{N} (\overline{f}(\mathbf{W}; \boldsymbol{x}_j) - y_j)^2, \tag{4}
$$

$$
\hat{l}_{\text{pretrained}} = \frac{1}{N} \sum_{j=1}^{N} (f(\mathbb{E}[\mathbf{W}]; \boldsymbol{x}_j) - y_j)^2. \tag{5}
$$

The above formulas are for a network with a scalar output. They can be easily extended to the vector output case by averaging over all outputs. In the LeNet example, we take the output of *softmax* layer to calculate squared losses. The noise is assumed *i.i.d.* Gaussian centered around zero with a fixed SNR $\sigma_{W,l}^2/\sigma_{N,l}^2$ in each layer $l$. The numerical values of the above losses are estimated using the entire test dataset for different noise levels. Results are shown in Figure 2(a). $\hat{l}_{bias}$ is initially equal to $\hat{l}_{pretrained}$ and $\hat{l}_{var} = 0$ when there is no noise. However, as noise level rises, they increase in magnitude and become much more important than $\hat{l}_{pretrained}$. $\hat{l}_{var}$ overtakes $\hat{l}_{bias}$ to become the predominant loss term in a noisy LeNet at $\sigma_N/\sigma_W \simeq 0.6$. It is useful to note that $l_{bias}$ increases with noise entirely due to nonlinearity in the network, which is ReLU in the case of LeNet. In a linear model, $l_{bias}$ should be equal to $l_{pretrained}$ as we would have $f(\mathbb{E}[\mathbf{W}]; \boldsymbol{x}) = \mathbb{E}[f(\mathbf{W}; \boldsymbol{x})]$. A model trained in a conventional manner is thus not optimized for the real loss it is going to encounter on a noisy accelerator. Special retraining is required to improve its noise tolerance. In Figure 2(a), we show how the model accuracy degrades with a rising noise level for the baseline LeNet and its deeper and wider variants. The deeper network is obtained by stacking two more convolutional layers of width 16 in front of the baseline network and the wider network is obtained by increasing the widths of each layer in the baseline to 128, 256, 2048 respectively. Performance degradation due to noise is worse for the deeper variant and less severe for the wider one. A more detailed discussion of the network architecture effect on its performance under noise is offered in Section 4.2

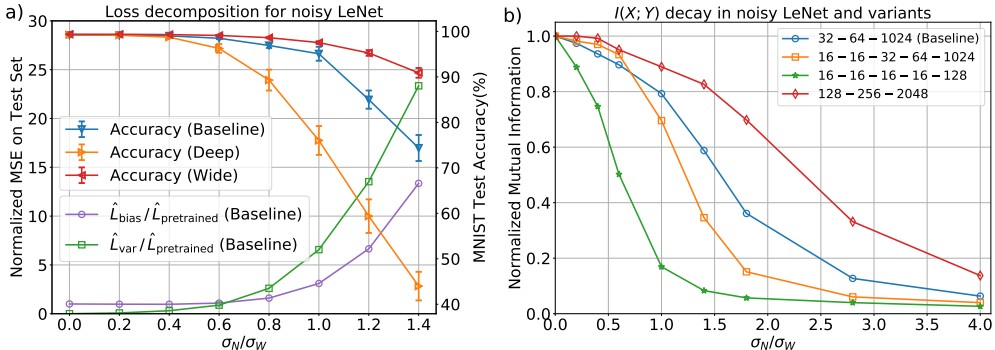

Figure 2: (a) Different loss terms on the test dataset and model test accuracy as a function of noise standard deviation, the losses are normalized to the pretrained model loss $\hat{l}_{pretrained}$, calculated using clean weights. Accuracy is calculated by performing the inference 100 times on the test set, error bars show the standard deviation.(b) Estimate of normalized mutual information between the input and output of the baseline LeNet and its variants as a function of noise standard deviation. A random subset of 200 training images are used for this estimate, with each inference repeated 100 times on a random realization of the network to estimate $H(Y|X)$. Mutual information decays with rising noise, deeper and narrower networks are more susceptible to this decay.

## 4.2 LOSS OF INFORMATION IN A NOISY NEURAL NETWORK

Information theory offers useful tools to study noise in neural networks. Mutual information $I(X;Y)$ characterizes the amount of information obtained on random variable $X$ by observing another random variable $Y$. The mutual information between $X$ and $Y$ can be related to Shannon entropy by

$$I(X;Y) = H(Y) - H(Y|X). \qquad (6)$$

Mutual information has been used to understand DNNs (Tishby & Zaslavsky, 2015; Saxe et al., 2018). Treating a noisy neural network as a noisy information channel, we can show how information about the input to the neural network diminishes as it propagates through the noisy computation. In this subsection, $X$ is the input to the neural network and $Y$ is the output. Mutual information is estimated for the baseline LeNet model and its variants using Equation 6. When there is no noise, the term $H(Y|X)$ is zero as $Y$ is deterministic once the input to the network $X$ is known, therefore $I(X;Y)$ is just $H(Y)$ in this case. Shannon entropy $H(Y)$ can be estimated using a standard discrete binning approach (Saxe et al., 2018). In our experiment, $Y$ is the output of the *softmax* layer

which is a vector of length 10. Entropy $H(Y)$ is estimated using four bins per coordinate of $Y$ by

$$\hat{H}(Y) = -\sum_{i=1}^{N} p_i \log(p_i), \tag{7}$$

where $p_i$ is the probability that an output falls in the bin $i$. When noise is introduced to the weights, the conditional entropy $H(Y|X)$ is estimated by fixing the input $X = x$ and performing multiple noisy inferences to calculate $\hat{H}(Y|X = x)$ with the above binning approach. $\hat{H}(Y|X = x)$ is then averaged over different input $x$ to obtain $\hat{H}(Y|X)$. This estimate is performed for LeNet and its variants with different noise levels. Results are shown in Figure 2(b). The values are normalized to the estimate of $I(X;Y)$ at zero noise. Mutual information between the input and the output decays towards zero with increasing noise in network weights. Furthermore, mutual information in a deeper and narrower network decays faster than in a shallower and wider network. Intuitively, information from the input undergoes more noisy compute when more layers are added to the network, while a wider network has more redundant paths for the information to flow, thus better preserving it. An information theoretic bound of mutual information decay as a function of network depth and width in a noisy neural network will be treated in our follow-up work. Overall, noise is damaging the learning capacity of the network. When the output of the model contains no information from its input, the network loses all ability to learn. For a noise level that is not so extreme, a significant amount of mutual information remains, which indicates that useful learning is possible even with a noisy model.

## 5   COMBINING NOISE INJECTION AND KNOWLEDGE DISTILLATION

### 5.1   METHODOLOGY

Noise injection during training is one way of exposing network training to a more realistic loss as randomly perturbing weights simulates what happens in a real noisy analog device, and forces the network to adapt to noise during training. Noise injection only happens in training during forward propagation, which can be considered as an approximation for calculating weight gradients with a straight-through-estimator (STE) (Bengio et al., 2013). At each forward pass, the weight $\mathbf{W}^l$ of layer $l$ is drawn from an *i.i.d.* Gaussian distribution $\mathcal{N}(\mathbf{W}^l; \mathbf{W}_0^l, \sigma_{N,l}^2 \mathbf{I})$. The noise is referenced to the range of representable weights $W_{\max}^l - W_{\min}^l$ in that particular layer

$$\sigma_{N,l} = \eta(W_{\max}^l - W_{\min}^l), \tag{8}$$

where $\eta$ is a coefficient characterizing the noise level. During back propagation, gradients are calculated with clean weights $\mathbf{W}_0^l$, and only $\mathbf{W}_0^l$ gets updated by applying the gradient. $W_{\max}^l$ and $W_{\min}^l$ are hyperparameters which can be chosen with information on the weight distributions.

Knowledge distillation was introduced by Hinton et al. (2015) as a way for training a smaller student model using a larger model as the teacher. For an input to the neural network $\boldsymbol{x}$, the teacher model generates logits $z_i^{\mathrm{T}}$, which are then turned into a probability vector by the *softmax* layer

$$q_i^{\mathrm{T}} = \sigma(z_i^{\mathrm{T}}; T) = \frac{\exp(z_i^{\mathrm{T}}/T)}{\sum_j \exp(z_j^{\mathrm{T}}/T)}. \tag{9}$$

The temperature, $T$, controls the softness of the probabilities. The teacher network can generate softer labels for the student network by raising the temperature $T$. We propose to use a noise free clean model as the teacher to train a noisy student network. The student network is trained with noise injection to match a mix of hard targets and soft targets generated by the teacher. Logits generated by the student network are denoted as $z_i^{\mathrm{S}}$. A loss function with distillation for the student model can be written as

$$\mathcal{L}(\boldsymbol{x}; \mathbf{W}^{\mathrm{S}}; T) = \mathcal{H}(\sigma(z_i^{\mathrm{S}}; T=1), y_{\mathrm{true}}) + \alpha T^2 \mathcal{H}(\sigma(z_i^{\mathrm{S}}; T), q_i^{\mathrm{T}}) + \mathcal{R}(\mathbf{W}_0^{\mathrm{S}}). \tag{10}$$

Here $\mathcal{H}$ is cross-entropy loss, $y_{\mathrm{true}}$ is the one-hot encoding of the ground truth, and $\mathcal{R}$ is the $L_2$-regularization term. Parameter $\alpha$ balances relative strength between hard and soft targets. We follow the original implementation in Hinton et al. (2015), which includes a $T^2$ factor in front of the soft target loss to balance gradients generated from different targets. The student model is then trained

with Gaussian noise injection using this distillation loss function. The vanilla noise injection training corresponds to the case where $\alpha = 0$. If the range of weights is not constrained and the noise reference is fixed, the network soon learns that the most effective way to decrease the loss is to increase the amplitude of the weights, which increases the effective SNR. There are two possible ways to deal with this problem. Firstly, the noise reference could be re-calculated after each weight update, thus updating the noise power. Secondly, we can constrain the range of weights by clipping them to the range $[W_{\min}^l, W_{\max}^l]$, and use a fixed noise model during training. We found that in general the second method of fixing the range of weights and training for a specific noise yields more stable training and better results. Therefore, this is the training method that we adopt in this paper. A schematic of our proposed method is shown in Figure 5 of the Appendix.

During training, a clean model is first trained to its full accuracy and then weight clipping is applied to clip weights in the range $[W_{\min}^l, W_{\max}^l]$. The specific range is chosen based on statistics of the weights. Fine-tuning is then applied to bring the weight-clipped clean model back to full accuracy. This model is then used as the teacher to generate soft targets. The noisy student network is initialized with the same weights as the teacher. This can be considered as a warm start to accelerate retraining. As we discussed earlier, the range of weights is fixed during training, and the noise injected into the student model is referenced to this range.

Our method also supports training for low precision noisy models. Quantization reflects finite precision conversion between analog and digital domains in an analog accelerator. Weights are uniformly quantized in the range $[W_{\min}^l, W_{\max}^l]$ before being exposed to noise. In a given layer, the input activations are quantized before being multiplied by noisy weights. The output results of the matrix multiplication are also quantized before adding biases and performing batch normalization, which are considered to happen in digital domain. When training with quantization, the straight-through-estimator is assumed when calculating gradients with back propagation.

## 5.2 EXPERIMENTAL RESULTS

In order to establish the effectiveness of our proposed method, experiments are performed for different networks and datasets. In this section we mainly focus on bigger datasets and models, while results on LeNet and its variants with some discussion of network architecture effect can be found in Figure 6 of the Appendix. ResNets are a family of convolutional neural networks proposed by He et al. (2016), which have gained great popularity in computer vision applications. In fact, many other deep neural networks also use ResNet-like cells as their building blocks. ResNets are often used as industry standard benchmark models to test hardware performance. The first set of experiments we present consist of a ResNet-32 model trained on the CIFAR10 dataset. In order to compare fairly with the previous work, we follow the implementation in Joshi et al. (2019), and consider a ResNet-32(v1) model on CIFAR10 with weight clipping in the range $[-2\sigma_{W,l}, 2\sigma_{W,l}]$. The teacher model is trained to an accuracy of $93.845\%$ using stochastic gradient descent with cosine learning rate decay (Loshchilov & Hutter, 2016), and an initial learning rate of $0.1$ (batch size is $128$). The network is then retrained with noise injection to make it robust against noise. Retraining takes place for $150$ epochs, the initial learning rate is $0.01$ and decays with the same cosine profile. We performed two sets of retraining, one without distillation in the loss ($\alpha = 0$), and another with distillation loss ($\alpha = 1$). Everything else was kept equal in these retraining runs. Five different noise levels are tested with five different values of $\eta$: $\{0.02, 0.04, 0.057, 0.073, 0.11\}$.

Results are shown in Figure 3(a). Every retraining run was performed twice and inference was performed 50 times on the test dataset for one model, to generate statistically significant results. Temperature was set to $T = 6$ for the runs with distillation. We found that an intermediate temperature between 2 and 10 produces better results. The pretrained model without any retraining performs very poorly at inference time when noise is present. Retraining with Gaussian noise injection can effectively recover some accuracy, which we confirm as reported in Joshi et al. (2019). Our method of combining noise injection with knowledge distillation from the clean model further improves noise resilience by about $40\%$ in terms of $\eta$, which is an improvement of almost $2\times$ in terms of noise power $\sigma_N^2$.

The actual noise level in a given device can only be estimated, and will vary from one device to another and even fluctuate depending on the physical environment in which it operates (Section 3). Therefore, it is important that any method to enhance noise robustness can tolerate a range of noise

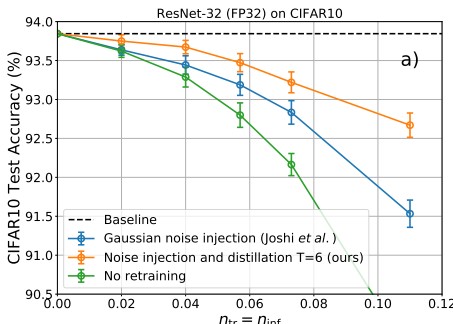 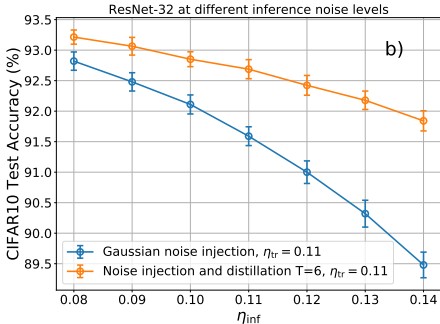

Figure 3: (a) Test accuracy as a function of noise level, here we have $\eta_{\mathrm{tr}} = \eta_{\mathrm{inf}}$, error bars show the standard deviation of different training and inference runs. Our method with distillation achieves the best robustness. (b) Comparison of model performance at noise levels different from the training level.

levels. Our method offers improved noise robustness, even when the actual noise at inference time is different from that injected at training time. It is shown in Figure 3(b) that the model obtained from distillation is more accurate and less sensitive to noise level differences between training and inference time. This holds for a range of different inference noise levels around the training level. In the previous experiments, we assume a fixed noise level parameterized by $\eta$. On real analog hardware, there could be additional non-idealities such as variation in noise level due to temperature fluctuation and nonuniform noise profile on different NVM cells due to statistical variation in the manufacturing process. We have conducted additional experiments to account for these effects.

Results from the experiments are shown in Table 1. Temporal fluctuation represents noise level variation over time. Noise $\eta$ is randomly sampled from $\mathcal{N}(\eta; \eta_0, \sigma_\eta^2)$ for each inference batch. A noise temporal fluctuation level of $10\%$ means that $\sigma_\eta = 0.1\eta_0$. Spatial noise level fluctuation introduces nonuniform diagonal terms in the noise covariance matrix. More concretely, each weight noise in our previous model is multiplied by a scale factor $\lambda_w$ with $\lambda_w$ drawn from a Gaussian distribution $\mathcal{N}(\lambda_w; 1, \sigma_\lambda^2)$. A noise spatial fluctuation level of $10\%$ means that $\sigma_\lambda = 0.1$. The scale factors are generated and then fixed when the network is instantiated, therefore the noise during network inference is non *i.i.d.* in this case. Results from our experiments show that there is no significant deviation when a combination of these non-ideal noise effects are taken into account.

Table 1: ResNet-32 on CIFAR10 with analog non-idealities: our method of combining distillation and noise injection consistently achieves the best accuracy under different analog non-ideal effects.

| Noise level | $\eta = 0.05$ | | | | |
|---|---|---|---|---|---|
| Non-ideal fluctuation type | Temporal 10% Spatial 0% | Temporal 20% Spatial 0% | Temporal 0% Spatial 10% | Temporal 0% Spatial 20% | Temporal 20% Spatial 20% |
| No retraining | 93% +/- 0.14% | 92.98% +/- 0.18% | 92.98% +/- 0.15% | 92.95% +/- 0.15% | 92.94% +/- 0.15% |
| Noise injection | 93.18% +/- 0.13% | 93.03% +/- 0.15% | 93.1% +/- 0.14% | 93.15% +/- 0.15% | 93.11% +/- 0.13% |
| Distillation and noise injection | **93.56%** **+/- 0.12%** | **93.55%** **+/- 0.11%** | **93.55%** **+/- 0.13%** | **93.51%** **+/- 0.12%** | **93.53%** **+/- 0.12%** |
| Noise level | $\eta = 0.1$ | | | | |
| No retraining | 90.46% +/- 0.19% | 90.22% +/- 0.27% | 90.5% +/- 0.2% | 90.4% +/- 0.23% | 90.1% +/- 0.3% |
| Noise injection | 91.87% +/- 0.17% | 91.93% +/- 0.2% | 91.91% +/- 0.2% | 91.79% +/- 0.18% | 91.81% +/- 0.17% |
| Distillation and noise injection | **92.83%** **+/- 0.18%** | **92.77%** **+/- 0.14%** | **92.88%** **+/- 0.14%** | **92.89%** **+/- 0.14%** | **92.86%** **+/- 0.15%** |

The performance of our training method is also validated with quantization. A ResNet-18(v2) model is trained with quantization to 4-bit precision (ENOB) for both weights and activations. This corresponds to 4-bit precision conversions between digital and analog domains. A subset of training

data is passed through the full precision model to calibrate the range for quantization – we choose the $0.1\%$ and $99.9\%$ percentiles as $q_{\min}$ and $q_{\max}$ for the quantizer. This range of quantization is fixed throughout training. The quantized model achieves an accuracy of $92.91\%$ on the test dataset when no noise is present. The model is then re-trained for noise robustness. The noise level is referenced to the range of quantization of weights in one particular layer, such that $W_{\min}^l = q_{\min,l}$ and $W_{\max}^l = q_{\max,l}$. Results are shown for the same set of $\eta$ values in Figure 4(a). In the distillation retraining runs, the full-precision clean model with an accuracy of $93.87\%$ is used as the teacher and temperature is set to $T = 6$. Due to extra loss in precision imposed by aggressive quantization, accuracy of the pretrained quantized model drops sharply with noise. At $\eta = 0.057$, the model accuracy drops to $87.5\%$ without retraining and further down to $80.9\%$ at $\eta = 0.073$. Even retraining with noise injection struggles, and the model retrained with only noise injection achieves an accuracy of $90.34\%$ at $\eta = 0.073$. Our method of combining noise injection and distillation stands out by keeping the accuracy loss within $1\%$ from the baseline up to a noise level of $\eta \simeq 0.07$.

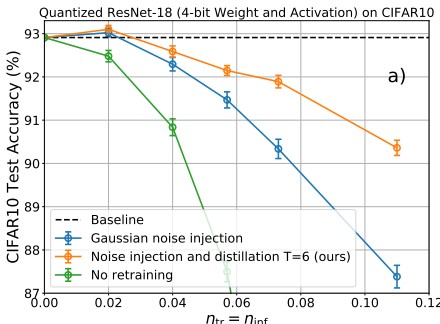 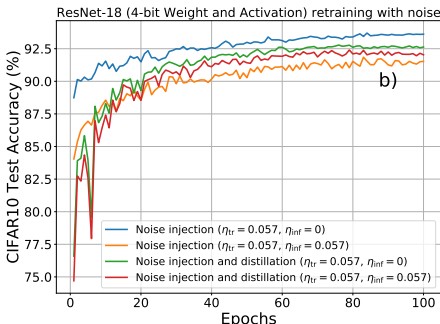

Figure 4: (a) Test accuracy as a function of noise level for 4-bit ResNet-18, here we have $\eta_{\mathrm{tr}} = \eta_{\mathrm{inf}}$, error bars show the standard deviation of different training and inference runs. Retraining with distillation and noise injection achieves the best results with quantization. (b) Test accuracy of different models during retraining with noise level $\eta = 0.057$.

One interesting aspect of using distillation loss during retraining with noise can be seen in Figure 4(b). The evolution of model accuracy on the test dataset is shown. When no distillation loss is used, the model suffers an accuracy drop (difference between blue and orange curves) around $2.08\%$ when tested with noise. The drop (difference between green and red curves) is significantly reduced to around $0.6\%$ when distillation loss is used. This observation indicates that training with distillation favors solutions that are less sensitive to noise. The final model obtained with distillation is actually slightly worse when there is no noise at inference time but becomes superior when noise is present.

Results on the ImageNet dataset for a ResNet-50(v1) network are shown in Table 2 to demonstrate that our proposed approach scales to a large-scale dataset and a deep model. A ResNet-50 model is first trained to an accuracy of $74.942\%$ with weight clipping in the range $[-2\sigma_{W,l}, 2\sigma_{W,l}]$. This range is fixed as the reference for added noise. For ResNet-50 on ImageNet, only three different noise levels are explored, and the accuracy degrades very quickly beyond the noise level $\eta = 0.06$, as the model and the task are considerably more complex. Retraining runs for 30 epochs with an initial learning rate of $0.001$ and cosine learning rate decay with a batch size of 32. For distillation, we used $\alpha = 1$ and $T = 6$ as in previous experiments. Results are collected for two independent training runs in each setting and 50 inference runs over the entire test dataset. The findings confirm that training with distillation and noise injection consistently delivers more noise robust models. The accuracy uplift benefit also markedly increases with noise.

## 6 DISCUSSION

**Effects of distillation** Knowledge distillation is a proven technique to transfer knowledge from a larger teacher model to a smaller, lower capacity student model. This paper shows, for the first time, that distillation is also an effective way to transfer knowledge between a clean model and its noisy

Table 2: ResNet-50 on ImageNet at different noise levels, showing the Top-1 accuracy on the test dataset, with no quantization applied. Uncertainty is the standard deviation of different training and inference runs.

| Noise level / Training method | $\eta = 0$ | $\eta = 0.02$ | $\eta = 0.04$ | $\eta = 0.06$ |
|---|---|---|---|---|
| No retraining | 74.942% | 72.975% +/- 0.095% | 64.382% +/- 0.121% | 46.284% +/- 0.179% |
| Gaussian noise injection | 74.942% | 73.513% +/- 0.091% | 70.142% +/- 0.129% | 65.285% +/- 0.168% |
| Distillation and noise injection | 74.942% | **74.005% +/- 0.096%** | **71.442% +/- 0.111%** | **67.525% +/- 0.162%** |

counterpart, with the novel approach of combining distillation with noise injection during training. We give some intuition for understanding this effect with the help of Section 4.2: a noisy neural network can be viewed as a model with reduced learning capacity by the loss of mutual information argument. Distillation is therefore acting to help reduce this capacity gap.

In our experiments, distillation shows great benefit in helping the network to converge to a good solution, even with a high level of noise injected in the forward propagation step. Here, we attempt to explain this effect by the reduced sensitivity of distillation loss. An influential work by Papernot et al. (2016) shows that distillation can be used to reduce the model sensitivity with respect to its input perturbations thus defending against some adversarial attacks. We argue that distillation can achieve a similar effect for the weights of the network. Taking the derivative of the $i$-th output of the student network $q_i^{\mathrm{S}}$ at temperature $T$ with respect to a weight $w$ yields

$$\frac{\partial q_i^{\mathrm{S}}}{\partial w} = \frac{1}{T} \frac{\exp(z_i/T)}{\left(\sum_j \exp(z_j/T)\right)^2} \sum_j \exp(z_j/T) \left(\frac{\partial z_i}{\partial w} - \frac{\partial z_j}{\partial w}\right). \tag{11}$$

The $1/T$ scaling makes the output less sensitive to weight perturbation at higher temperature, thus potentially stabilizing the training when noise is injected into weights during forward propagation. We plan to work on a more formal analysis of this argument in our future work.

**Hardware Performance Benefits** The improvements in noise tolerance of neural networks demonstrated in this work have a potential impact on the design of practical analog hardware accelerators for neural network inference. Increased robustness to noisy computation at the model training level potentially means that the specification of the analog hardware can be relaxed. In turn, this can make it easier to achieve the hardware specification, or even allow optimizations to further reduce the energy consumption. An in-depth discussion of the trade-off between compute noise performance and hardware energy dissipation is beyond the scope of this paper, but we refer the interested reader to Rekhi et al. (2019) for more details. In summary, we believe that machine learning research will be a key enabler for practical analog hardware accelerators.

## 7 CONCLUSION

Analog hardware holds the potential to significantly reduce the latency and energy consumption of neural network inference. However, analog hardware is imprecise and introduces noise during computation that limits accuracy in practice. This paper explored the training of noisy neural networks, which suffer from reduced capacity leading to accuracy loss. We propose a training methodology that trains neural networks via distillation and noise injection to increase the accuracy of models under noisy computation. Experimental results across a range of models and datasets, including ImageNet, demonstrate that this approach can almost double the network noise tolerance compared with the previous best reported values, without any changes to the model itself beyond the training method. With these improvements in the accuracy of noisy neural networks, we hope to enable the implementation of analog inference hardware in the near future.

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

# A  APPENDIX

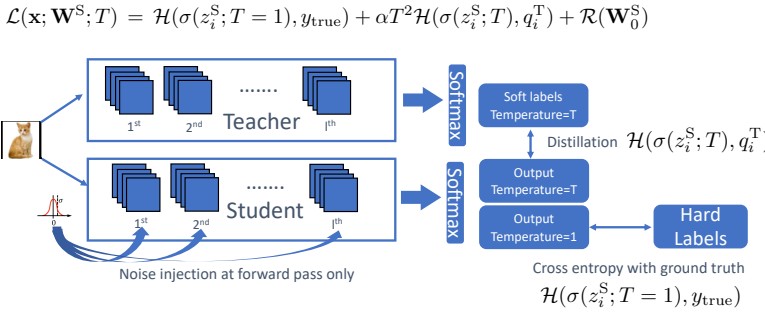

$$\mathcal{L}(\mathbf{x}; \mathbf{W}^{S}; T) = \mathcal{H}(\sigma(z_i^{S}; T=1), y_{\text{true}}) + \alpha T^2 \mathcal{H}(\sigma(z_i^{S}; T), q_i^{T}) + \mathcal{R}(\mathbf{W}_0^{S})$$

Figure 5: Schematic of our retraining method combining distillation and noise injection.

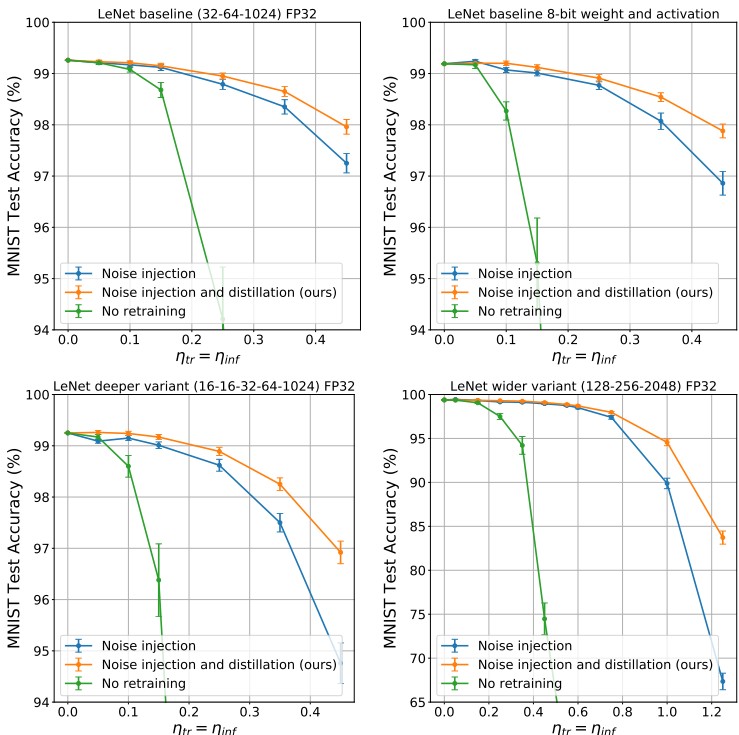

Figure 6: Results on LeNet and its variants show that our method of combining distillation and noise injection improves noise robustness for different model architectures on MNIST. The benefit of our method is the most significant when the network struggles to learn with vanilla noise injection retraining method. This threshold noise level depends on the network architecture, as we have remarked for mutual information decay.

