# OpenReview forum: "Noisy Machines: Understanding noisy neural networks and enhancing robustness to analog hardware errors using distillation"
_ICLR.cc/2020/Conference — Reject_

### Official Review · AnonReviewer1 · 2019-10-19
**Official Blind Review #1**

**Rating:** 6

**Review:**

* Summary *
The article on "Noisy Machines" addresses the issue of implementing deep neural network inference on a noisy hardware computing substrate, e.g. analog accelerators. This is an important topic because analog devices allow fast and energy efficient inference, which is crucial for inference at the edge. Because of their analog nature such devices suffer from noisy computations, and in this article the case of noisy weights is studied.

The main contributions of this article are the following:
- an analysis of the performance loss in noisy networks by means of information theory and empirical results
- the idea of combining noise injection during training with knowledge distillation
- experimental evidence for a LeNet5 on MNIST, CIFAR10, and ResNet-50 on ImageNet

It has been shown in the literature that noise injection during training is an effective way to increase the noise robustness of neural networks. Relevant literature in this domain is cited. The novelty of the approach is to combine noise injection with distillation, by using the noise-free network as a teacher for the noisy network, which is initialized with the weights of the teacher. This is a novel variant of distillation and sounds like a simple to implement trick with beneficial results for increasing noise resiliency of networks. It is also proposed and shown that the method works for quantized networks.

The experimental results show that the combination of distillation and noise injection outperforms pure noise injection on all networks, as well as noisy inference without retraining. The effect is even more pronounced for quantized networks.

* Evaluation *
Overall I like this paper and think it is suitable to accept for ICLR, because it addresses an important practical problem of implementing deep networks on efficient hardware. The paper is well written and simple to understand and should be easy to implement (it would really help here providing code for the examples though). To the best of my knowledge I have not seen precisely this combination of noise injection and distillation, although there is a lot of literature about each individual approach. I appreciate that the authors made an effort to not just show empirical results but also motivate their findings by theory, although the argumentation stays a bit superficial.

What I am mainly missing are two points:
1. The assumed noise model of i.i.D. Gaussian weights is the simplest possible, and might deviate quite a bit in actual analog hardware. I would have liked to see a noise model that is derived from actual hardware observations, or maybe even a prototype implementation in hardware, such as was done e.g. in Binas et al. 2016. At the very least I would suggest to test the model on other noise models, including temporally changing noise levels, which could be a realistic scenario due to temperature fluctuations or other events.

2. The experimental results focus on MNIST, CIFAR10, and later briefly on ImageNet. While the results are quite convincing on MNIST and CIFAR, these are easier datasets with usually well separable classes, so the effect of noisy inference might not be as pronounced, as in datasets with more confusion even in the clean case. In the case of ImageNet (Table 1) it looks like the difference to pure noise injection is not as big as it was in the CIFAR case, but here also only lower noise levels were tested. I would recommend testing also the same noise range as for CIFAR to understand whether distillation always shows the desired benefits, or if this is a diminishing effect for larger networks. Overall it would help to understand how the effect scales with network depth, e.g. by comparing the information loss for different ResNet depths.

I'm giving weak accept and would change to accept if there could be clarification on how the approach scales to different network architectures and noise models closer to actual hardware. I also recommend publishing some example code for this approach.

**Experience Assessment:**

I have published in this field for several years.

**Review Assessment: Checking Correctness Of Derivations And Theory:**

I carefully checked the derivations and theory.

**Review Assessment: Checking Correctness Of Experiments:**

I carefully checked the experiments.

**Review Assessment: Thoroughness In Paper Reading:**

I read the paper at least twice and used my best judgement in assessing the paper.

---

> ### Author Response · Authors · 2019-11-15
> **Author response to Reviewer #1**
>
> Author Response:
>
> Many thanks for taking the time to review our paper and the constructive comments. We've added additional experimental results and reworked the relevant sections of the paper in response to the reviewer feedback.
>
> - Q1 Noise model
> We believe that the simple Gaussian noise model used in the paper is a good approach to modeling real analog hardware systems, which has been confirmed previously in "Accurate deep neural network inference using computational phase-change memory"(https://arxiv.org/abs/1906.03138)  by Joshi et al. Their experiments show (Fig 3c in Joshi et al.) that very similar measured accuracy can be achieved when weights trained with the Gaussian noise model are transferred to the actual hardware. And in fact, we are already considering modeling of key circuit blocks such as digital-to-analog and analog-to-digital converters within our formulation of weight and activation quantization, which we've emphasized in Section 3, paragraph 2.
>
> Nonetheless, we do agree that robustness to spatially-varying or even time-varying noise models is a very interesting question. Therefore we have expanded the results in this regard by adding experiments with time-varying (temperature fluctuation) and spatially-varying (manufacturing process variation) noise models in Table 1 of Section 5.2 to broaden this aspect of the work.
>
> In summary, we have found no significant deviation in these new experiments. The improvement from our proposed retraining method is quite consistent. We would like to thank the reviewer for suggesting these experiments which have strengthened the results of our paper. In addition to the new experimental results, we've also extended the discussion of some of the general challenges and considerations in modeling analog hardware non-idealities (Section 3).
>
> - Q2 Scaling of results over different network architectures
> We also think this is an interesting question, so we expanded our results to provide some data. Following the reviewer's suggestion, we have added further experiments in Section 4.2 (Figure 2) to better understand the mutual information decay over both wider and deeper neural network architectures. We have also added more results in the Appendix (Figure 6) showing how our method compares to pure noise injection on different network architectures. Our results suggest that the improvement does not diminish for larger networks given the same dataset. Larger networks can be either more susceptible to noise or less so, depending on their architectures (see discussion in Section 4.2). To obtain the same amount of uplift from our method, one may need to go to lower or higher noise levels depending on the architectures.
>
> The reviewer's observation that the benefit is less pronounced on ImageNet than on CIFAR10 is likely due to the nature of the dataset itself. As the reviewer has pointed out, ImageNet is a more difficult task, even without noise. The reason we reported relatively low noise levels on ImageNet experiments is because the accuracy degradation is already quite significant compared to the CIFAR10 cases. From a practical application point of view, pushing to even higher noise levels is arguably less interesting. Nevertheless, we understand the request for higher noise levels on ImageNet experiments. We are currently running more experiments on ImageNet. Given the short time-frame of rebuttal, we don't have the results quite yet (also more training epochs are needed for higher noise levels). We will incorporate the results in our paper as they come out.
>
> - Q3 Open sourcing code examples
> Finally, we do plan to open source code used in the paper.
> We are currently working through an internal legal approval process to do so, but are confident we will be able to have it online before the conference.

---

### Official Review · AnonReviewer2 · 2019-10-23
**Official Blind Review #2**

**Rating:** 6

**Review:**

The authors of the manuscript study how inherent noise in the analog neural networks affects its accuracy. This is a very important topic as neural network based inference becomes uobiqutous and is required to run with very low power consumption and latency.

The manuscipt considers a system where the values of the neural network weights and biases are experiencing i.i.d Gaussian noise, which is a pretty good assumptions. However, in heavy use the system may warm up, and then there could be an effect that is correlated accross different weights. The noise model used would not be able to ensure proper inference in these conditions. I would like to see a discussion on the effect of correlations in the injected noise.

The mutual information is considered and evaluated for a the "noisy" and "clean" versions and the result is according to expectations. The some degree, I do not see this part very valuable, as it does not bring any particular insights on the analog neural network operation. Rather I woudl like to see how, the analog performance under noise scales as the neural has  more layers.  Also, the noise behavior of an analog RNN woudl be very interesting.

The authors have detected that even, if the neural network trainded without noise is not robust when the weigths fluctuate, the trained network is a good starting point for transfer learning. To some degree I do not find this to be a very inventive step as transfer learning has shown to able to cross much larger training data set alterations.

Good solid work, but lacking non-obvious results, and I do not see manuscript adressing the the harder challenges. However, the quantified results may have a notable practical importance.




**Experience Assessment:**

I have read many papers in this area.

**Review Assessment: Checking Correctness Of Derivations And Theory:**

I assessed the sensibility of the derivations and theory.

**Review Assessment: Checking Correctness Of Experiments:**

I assessed the sensibility of the experiments.

**Review Assessment: Thoroughness In Paper Reading:**

I read the paper thoroughly.

---

> ### Author Response · Authors · 2019-11-15
> **Author response to Reviewer #2**
>
> Author Response:
>
> Many thanks for taking the time to review our paper and the constructive comments. We've added additional experimental results and reworked the relevant sections of the paper in response to the reviewer feedback.
>
> - Q1 Correlated noise models
> The question of changes in the noise model as the system warms up is a good one and another reviewer was also interested in this. It is our understanding that system warming up is less likely to result in statistical correlation between weight noise but rather spatially inhomogeneous noise levels. For example, if the center of an analog chip heats up more than its edges, the NVM cells at the center may have a higher noise level than the ones close to the edges.
>
> We have expanded the discussion of these non-idealities in Section 3, and we have also expanded the results in this regard by adding experiments with time-varying (temperature or supply voltage fluctuation) and spatially-varying (manufacturing process variation or temperature inhomogeneity due to warming up) noise models in Table 1 of Section 5.2 to broaden this aspect of the work. The spatially-varying noise model is effective non i.i.d.
>
> In summary, we have found no significant deviation in these new experiments. The improvement from our proposed retraining method remains quite consistent. Nonetheless, we would like to thank the reviewer for suggesting these experiments which have strengthened the results of our paper.
>
> - Q2 Analog performance scaling with network depth
> We also thought this was interesting and, again, another reviewer (#1) asked about this too.
> Following the reviewer's suggestion, we have added further experiments in Section 4.2 (Figure 2) to better understand the mutual information / model accuracy decay over both wider and deeper neural network architectures. We have also added more results in the Appendix (Figure 6) showing how our method compares to pure noise injection on different network architectures. We're grateful for the feedback, as these additional experiments have increased our understanding of the impact of analog noise on neural network performance.
> We are also interested in the performance of analog RNNs, but we feel this is beyond the scope of this paper, but we are planning to look at this for future work.
>
> - Q3 Lacking non-obvious results
> Finally, regarding non-obvious results, we think that this paper is a first step in starting to understand the behavior of analog neural networks. As other reviewers have remarked, this is a new area for the ML community, and we think that many are probably not aware of the interest around analog hardware for neural networks. Therefore, this paper serves an important role, not only in introducing the ML community to the challenges of designing and training models that are robust to analog noise, but at the same time outlining an approach that achieves state-of-the-art results. Additionally, our retraining approach of combining noise injection and distillation is indeed novel as other reviewers have remarked. We have also broadened the experiments around the noise model, providing results to support our choice of noise modeling. We believe this line of work could really make deployment of analog neural network hardware possible and hope to motivate other ML work in this direction. In that regard, we hope this paper can be a first step in building understanding and interest in the topic amongst the broader ML community.

---

### Official Review · AnonReviewer4 · 2019-11-04
**Official Blind Review #4**

**Rating:** 3

**Review:**

The manuscript illustrates how a noisy neural network can reduce the learning capacity. To mitigate this loss, the authors propose a method that combines the method of "noise injection and "knowledge distillation". However, from a conceptual point of view,  their contribution (i.e. (10) in Section 5,) is unclear to me. Specifically,  the authors are not precise about how do they merge the aforementioned previous ideas and come up with the new loss function (10).

Minor comment: Please correct (7).




**Experience Assessment:**

I do not know much about this area.

**Review Assessment: Checking Correctness Of Derivations And Theory:**

I did not assess the derivations or theory.

**Review Assessment: Checking Correctness Of Experiments:**

I did not assess the experiments.

**Review Assessment: Thoroughness In Paper Reading:**

I made a quick assessment of this paper.

---

> ### Author Response · Authors · 2019-11-15
> **Author response to Reviewer #4**
>
> Many thanks for taking the time to review our paper.
>
> We note that, in the reviewer's own words, this review should be considered a "quick assessment".
>
> Regarding (10), (10) is the usual loss function for distillation. The novelty of our work is in combining knowledge distillation and noise injection as other reviewers have remarked. We have included a schematic of our retraining method in Figure 5 of the Appendix to make things more clear.
>
> Thanks for catching the typo in (7), we have corrected it.

---

### Official Review · AnonReviewer3 · 2019-11-10
**Official Blind Review #3**

**Rating:** 3

**Review:**

I started reading the paper with high hopes. The abstract and the introduction were set up nicely and I was quite intrigued to see a theoretical analysis and practical implementation of a neural network using analogue hardware. However, as I read through the paper carefully multiple times, I realized that the expository opening description fails to live up to the standard it sets.

To be more specific, I did not enjoy the elaborate description of noisy analogue conductances as the noise models analyzed in the paper are not tested on any of such devices. The noise model introduced is fairly simplistic and arguably, the real-world systems are much more complex compared to such simplistic assumptions. The authors could have presented the paper as an analysis of knowledge distillation in neural network training. Even if the paper were presented that way, I would have doubted its chance of acceptance due to the incrementality of the theoretical contribution.

All in all, I believe this is a very promising direction to invest, but the paper is not quite ready for ICLR.

**Experience Assessment:**

I have read many papers in this area.

**Review Assessment: Checking Correctness Of Derivations And Theory:**

I assessed the sensibility of the derivations and theory.

**Review Assessment: Checking Correctness Of Experiments:**

I assessed the sensibility of the experiments.

**Review Assessment: Thoroughness In Paper Reading:**

I read the paper thoroughly.

---

> ### Author Response · Authors · 2019-11-15
> **Author response to Reviewer #3**
>
> Many thanks for taking the time to review our paper and the constructive comments.
>
> - Q1 Noise models
> We believe that the simple Gaussian noise model used in the paper is a good approach to modeling real analog hardware systems, which has been confirmed previously in "Accurate deep neural network inference using computational phase-change memory"(https://arxiv.org/abs/1906.03138)  by Joshi et al. Their experiments show (Fig 3c in Joshi et al.) that very similar measured accuracy can be achieved when weights trained with the Gaussian noise model are transferred to the actual hardware. And in fact, we are already considering modeling of key circuit blocks such as digital-to-analog and analog-to-digital converters within our formulation of weight and activation quantization, which we've emphasized in Section 3, paragraph 2.
>
> We have also expanded the results by adding experiments with time-varying (temperature fluctuation) and spatially-varying (manufacturing process variation) noise models in Table 1 of Section 5.2 to broaden this aspect of the work. In summary, we have found no significant deviation in these new experiments. The improvement from our proposed retraining method is quite consistent.
>
> - Q2 Practical implementation
> Sadly, there is currently no publicly available hardware to test our models on. Instead of waiting for practical analog hardware to become available, we think that research in the ML community can advance the practicality of analog neural networks. We've also noticed that this topic falls between the ML and hardware communities, but think that the ML community can contribute significantly in this area. As other reviewers have remarked, this is a new area for the ML community and many are probably unaware of the interesting ML challenges posed by analog neural networks. We hope to motivate other ML work in this direction. Analog deep learning hardware is a complex system level challenge and requires a system level solution. We don't claim to solve all the challenges in this one paper but rather we have carved out a well-defined ML problem from the broader challenge and presented a novel solution to it, we hope this paper can be a first step in building understanding and interest in the topic amongst the broader ML community.

---

### Public Comment · ~Micah_Goldblum1 · 2019-11-08
**An Interesting Connection**

Hi Authors,
Thank you for your interesting paper.  I noticed that your work concerning robustness via distillation is related to our work on producing adversarially robust networks using a variant of distillation.[1]  Please consider mentioning the relationship with our work in your next version.

[1] Goldblum, Micah, et al. "Adversarially Robust Distillation." arXiv preprint arXiv:1905.09747 (2019).

---

> ### Author Response · Authors · 2019-11-15
> **Thanks for pointing us to the paper**
>
> Thanks for pointing us to the paper, it's a very interesting read. We have mentioned it in our current version.

---

### Decision · Program_Chairs · 2019-12-19

**Decision:**

Reject

**Comment:**

This paper argues that NNs deployed to hardware needs to robust to additive noise and introduces two methods to achieve this.

The reviewers liked aspects of the paper and the paper is borderline. However, all in all sufficient reservations were raised to put the paper below the threshold. The criticism was constructive and can be used in an updated version submitted to next conference.

Rejection is recommended.